# Physical Activity Protocols in Non-Alcoholic Fatty Liver Disease Management: A Systematic Review of Randomized Clinical Trials and Animal Models

**DOI:** 10.3390/healthcare11141992

**Published:** 2023-07-10

**Authors:** Elisa Barrón-Cabrera, Raúl Soria-Rodríguez, Fernando Amador-Lara, Erika Martínez-López

**Affiliations:** 1Faculty of Nutrition and Gastronomy Sciences, Autonomous University of Sinaloa, Culiacan 80010, Mexico; elisabarron@uas.edu.mx; 2Program in Physical Activity and Lifestyle, University Center of Health Sciences, University of Guadalajara, Guadalajara 44100, Mexico; 3Department of Medical Clinics, University Center of Health Sciences, University of Guadalajara, Guadalajara 44100, Mexico; 4Institute of Translational Nutrigenetics and Nutrigenomics, Department of Molecular and Genomic Biology, University Center of Health Sciences, University of Guadalajara, Guadalajara 44100, Mexico

**Keywords:** physical activity, exercise training, liver diseases, fatty liver

## Abstract

Non-alcoholic fatty liver disease (NAFLD) is closely associated with other metabolic disease and cardiovascular disease. Regular exercise reduces hepatic fat content and could be the first-line treatment in the management of NAFLD. This review aims to summarize the current evidence of the beneficial effects of exercise training and identify the molecular pathways involved in the response to exercise to define their role in the resolution of NAFLD both in animal and human studies. According to the inclusion criteria, 43 animal studies and 14 RCTs were included in this systematic review. Several exercise modalities were demonstrated to have a positive effect on liver function. Physical activity showed a strong association with improvement in inflammation, and reduction in steatohepatitis and fibrosis in experimental models. Furthermore, both aerobic and resistance exercise in human studies were demonstrated to reduce liver fat, and to improve insulin resistance and blood lipids, regardless of weight loss, although aerobic exercises may be more effective. Resistance exercise is more feasible for patients with NAFLD with poor cardiorespiratory fitness. More effort and awareness should be dedicated to encouraging NAFLD patients to adopt an active lifestyle and benefit from it its effects in order to reduce this growing public health problem.

## 1. Introduction

In recent years, the prevalence of obesity has increased along with non-alcoholic fatty liver disease (NAFLD) and metabolic syndrome. Up to 25% of subjects with NAFLD may develop non-alcoholic steatohepatitis (NASH), which is characterized by the accumulation of triglycerides in the liver, inflammation, tissue injury, and apoptosis of hepatocytes, leading to liver cirrhosis and its complications, hepatic decompensation and hepatocellular carcinoma (HCC) [1]. In addition to liver-related mortality, NAFLD is associated with increased mortality associated mostly with cardiovascular disease, diabetes mellitus, and non-liver cancer [2]. The worldwide prevalence of NAFLD is estimated at around 25%, with a higher number of cases reported in South America and the Middle East [3], and is most frequent in men between 40 and 50 years old and women between 60 and 69 years old [4]. Moreover, being overweight during the early life stage is associated with a higher risk of developing NAFLD in adulthood [5]. Notably, 7–20% of subjects with NAFLD exhibit a lean phenotype [6].

Currently, no drug has been approved by international or local regulatory agencies specifically to treat NAFLD. However, there are many drugs approved to treat other components of metabolic syndrome, such as diabetes mellitus, dyslipidemia, or obesity. Instead, treatment has been focused on lifestyle modifications addressed to achieve weight loss through dietary control and increased physical activity [7]. The American Gastroenterological Association (AGA) advises patients with fatty liver to reduce over 5% of total body weight (TBW) to decrease hepatic steatosis, over 7% for inflammation resolution, and over 10% to resolve/stabilize fibrosis, as has been shown in several studies [8,9,10,11]. Although liver biopsy is the gold standard for diagnosis and staging of this disease, noninvasive tests are also used to investigate the presence of NAFLD and associated fibrosis, such as serum biomarkers, ultrasound analysis, elastography computed tomography, and proton magnetic resonance spectroscopy (^1^H-MRS) [12].

Studies in animals and humans support the theory that hepatocellular injury, present in NAFLD, is induced by an excess of primary metabolic substrates, i.e., glucose, fructose, and fatty acids, associated with obesity and insulin resistance in the liver [13].

A sedentary lifestyle has been linked to a detrimental prognosis in individuals with NAFLD [14]. Among individuals with obesity, people without physical activity exhibit elevated hepatic levels of free fatty acids compared to their physically active counterparts [15]. Consequently, engaging in structured and repetitive physical activities, characterized by careful consideration of factors such as intensity, frequency, duration, and personal preferences, may serve to mitigate the progression of NAFLD. In this context, lifestyle modifications, such as increased physical activity and exercise training, have a nominal low-cost compared to long drug-based treatment in the natural history of management of this clinical condition [14,15]. Additionally, it improves other metabolic risk factors, such as insulin resistance, dyslipidemia, obesity, type-2 diabetes mellitus, elevated blood pressure, and cardiovascular disease in patients with NAFLD [16].

There is strong evidence of the benefits of physical activity, along with restriction of caloric intake, as a primary approach in the management of NAFLD. However, the exercise prescription in terms of type, intensity, frequency and duration has not been standardized [8,17,18]. The American College of Sports Medicine (ACSM) recommends at least 150 min/week of moderate or 75 min/week of vigorous-intensity physical activity for all patients with NAFLD [19]. It is worth emphasizing that the objective of the present review is to provide updated information on various exercise program modalities, as well as to shed light on the key molecular pathways that play a significant role in NAFLD resolution, by drawing insights from both animal and human studies to achieve a specific exercise recommendation for NAFLD amelioration in humans supported by basic science.

## 2. Materials and Methods

This systematic review was designed and performed following the Preferred Reporting Items for Systematic Reviews and Meta-Analyses (PRISMA) statement [20]. All the included articles were organized according to whether the study design involved experimental models or human studies. The specific inclusion and exclusion criteria for the experimental models are described in the flow diagram in Figure 1a, while the criteria for human studies are described in Figure 1b.

### 2.1. Eligibility Criteria

We performed an electronic search of the English language literature using PubMed, EBSCO, and Google Scholar databases to identify articles that studied physical activity or exercise programs in the management of NAFLD in adults (>18 years) and animal models. The search followed the keywords used either alone or combined according to the PubMed MeSH database: “Nonalcoholic fatty liver disease”, “NAFLD”, “non-alcoholic steatohepatitis”, “NASH”, “obesity”, “exercise”, “exercise training”, “physical activity”, “aerobic training” and, “resistance training”. The search was restricted to studies published from January 2017 to November 2022 to include the most up-to-date studies on this topic.

### 2.2. Inclusion Criteria

For animal studies, we included studies that: (1) measured biomarkers or molecular pathways associated with NAFLD resolution in response to any type of exercise program training, (2) were published in the last 5 years, and (3) were performed in an animal model. For human studies, only randomized controlled trials (RCT) with supervised exercise training that aimed to assess the therapeutic effects of exercise on hepatic steatosis and/or metabolic effects in patients with NAFLD were included.

### 2.3. Exclusion Criteria

In the animal studies, titles and abstracts were screened to exclude the following: reviews; studies without reported details of exercise training protocols, such as time, intensity, duration, or type of exercise; studies combining exercise programs with other interventions, such as diet supplement, nutritional strategies, or drugs administration; studies focusing on different outcomes, such as insulin sensibility, cancer, lipids profile, prothrombotic state; studies of molecular pathways not involved in the resolution of NAFLD; and studies published in languages other than English. For human studies, the use of supplements or drugs as part of the intervention to manage NAFLD, as well as studies with insufficient information on the characteristics of the exercise (type, intensity, frequency, duration per session, and duration of the intervention), were excluded; studies including subjects younger than 18 years old, non-full-text articles, and non-English language studies were also excluded.

### 2.4. Quality Assessment of the RCT Included Studies

The quality of the RCT included was evaluated according to the revised Cochrane risk-of-bias tool for randomized studies (RoB 2) [21]. Two authors (FAL and RSR) independently evaluated the quality of trials, and discrepancies were resolved through discussion with a third reviewer (EML) to reach a consensus.

## 3. Results

We found a total of 15,925 articles in the initial comprehensive database search. Out of these, 5225 corresponded to experimental studies and 10,700 to human clinical studies. After removing duplicate articles and excluding articles that did not meet the inclusion criteria, 43 animal studies and 14 RCTs were included in this review. Table 1 summarizes the most recent studies regarding exercise training in NAFLD experimental models. Table 2 displays details of the studies about the human response mechanisms to regular exercise in NAFLD, including the exercise protocol training type, intensity, frequency, duration per session, duration of the intervention, as well as the main changes associated with NAFLD.

**Table 1 healthcare-11-01992-t001:** The effects of exercise training on experimental models with non-alcoholic fatty liver disease.

Animal Models	Intervention		Outcomes		Reference
	Exercise Protocol	Comparison Groups	Molecular Outcomes	Outcomes in NAFLD	
Inhibition of Lipid Accumulation Pathways					
Male C57BL/6J mice 5 weeks old n = 12/group	HIIT protocol: 2 min of running, followed by 2 min of rest for a total of 60 min. 3 d/week with one or two days of rest between each exercise session.	(1) Sedentary mice (2) HIIT training (3) Moderate-intensity continuous training (MIT)	↓ *Pparg*, *Dgat1*, *Acaca*, and *Acacb* gene expression	HIIT was superior to MIT to reduce adiposity, improve whole-body glucose tolerance, and ameliorate liver steatosis, inflammation, and fibrosis.	Fredrickson, et al., 2021 [22]
Male C57BL/6 mice 4 weeks old n = 40	Aerobic treadmill exercise: 50 min at 65–70% of VO_2_ max, 5 d/week for 8 weeks. Ladder climbing exercise: 65–70% of the maximum weight was attached to the mouse’s tail (lead) to climb the ladder, 5 d/week for 8 weeks.	(1) HFD + sedentary (2) HFD + aerobic exercise (3) HFD + resistance exercise	↑ Liver *Cpt1*, *p-AMPK*, and p-*AMPK*/*AMPK* ↓ Liver SREBP-1	Fat mass and liver triglycerides were significantly decreased.	Bae, et al., 2020 [23]
Male Wistar rats 8 weeks old n = 40	Continuous endurance training (CET): 5 min of running at 30–40% VO_2_ max for the warm-up period followed by 30 min of running at 60–65% VO_2_ max, and a 5 min cool down period of running at 30–40% VO_2_ max for each session. High-intensity interval training (HIIT): 5 min warm-up period of running at 30–40% VO_2_ max, followed by 5 cycles of alternating high-intensity and low-intensity intervals (2 min at 85–90% VO_2_ max and 2 min at 30–40% VO_2_ max), and a final 3 min cooling down period of running at 30–40% VO_2_ max during 8 weeks.	(1) Non-diabetic (2) HFHFD (3) HFHFD + CET (4) HFHFD + HIIT9	HIIT ↓ *Fas*, *Acc*, and *SREBP-1c* expression compared with HFHFD. Additionally, HIIT ↑ miR-122 expression was similar to the CET group.	Both HIIT and CET exercise attenuated hepatic steatosis; however, HIIT was more effective at reducing vesicular steatosis.	Kalaki-Jouybari, et al., 2020 [24]
Male wild-type (WT) and GCN2 knockout (GCN2KO) mice 6–8 weeks old n = 40	Mixed exercise protocol (combined aerobic and resistance exercises): Running at 10–12 m/min for 60 min at an inclination of 0° followed by executed ladder climbing at a 90° inclination with no tail loading for a total of 4 repetitions/set and 4 sets/day for 8 weeks.	(1) Sedentary WT HFD (2) Exercised WT HFD-fed mice (3) Sedentary GCN2 knock-out HFD-fed mice (4) Exercised GCN2 knockout HFD-fed mice	Exercise GCN2KO mice: ↓ *p-eIF2α* and *Atf4* expressions ↑ *p-Ampk*, *Sirt1*, and *Ppar-α* expressions	Improvement in hepatic steatosis and control of glucose intolerance.	Luo, et al., 2020 [25]
Male C57BL/6 mice 8 weeks old n = 49	Aerobic exercise: 40 min of treadmill, 5 d/week for 8 weeks.	(1) HFD + sedentary (2) HFD + training (3) Normal-diet + sedentary (4) Normal-diet + training	Exercise + diet and diet alone: ↓ *Cb1* and *Ampk* expressions ↑ *p-Ampk* and *Cpt1* expressions	Improvement in glucose tolerance and decrease of liver fat.	Ok, et al., 2018 [26]
Male C57BL/6J mice 8 weeks old n = 40	Aerobic exercise: 6 d/week, 18 m/min for 50 min, 6% slope for 8 weeks.	(1) Normal control (2) Normal aerobic exercise (3) HFD (4) HFD + aerobic exercise	↓ *Sra* expression and JNK/P38 signaling were inhibited ↑ *Atgl* expression	Improvement in hepatic steatosis, TC, LDL-C, and liver TG levels.	Wu, et al., 2022 [27]
Male C57BL/6J mice 8 weeks old n = 21	Aerobic exercise: 60 min of swim training, 5 d/week for 16 weeks.	(1) Normal diet (2) HFD (3) HFD + swim training	↑ AMPK/SIRT1 signaling, autophagy marker and *Cpt1* gene expression	Reduction in liver steatosis and insulin resistance; activation of lipophagy.	H. Li, et al., 2021 [28]
Male C57BL/6 mice 5–6 weeks old n = 50	Endurance training: The mice ran for 26 min a day, spending 1 min at 6 m/min, 1 min at 8 m/min, 22 min at 10 m/min, and 2 min 12 m/min, 5 d/week for 8 weeks (average to 70% of their VO_2_ max).	(1) Control diet (2) Control diet + endurance-trained (3) HFD and sedentary (4) HFD + endurance-trained (5) HFD and then changed to control diet + endurance-trained	↓ *Srebf1*, *Srebf2*, *Cptia*, *Ppar-γ* and *Cd36* expression ↑ *Ppar-α*, *Ppar-δ*, and *Cyp4a-10* were enhanced	Reduction in the size of liver fat droplets, reversion of HFD-induced steatosis, and a significant 30% decrease in fibrosis.	Melo, et al., 2021 [29]
Male Wistar rats 6 weeks old n = 37	Aerobic and resistance exercise: moderate intensity by 8 weeks of running on a treadmill and climbing a ladder for 5 d/week.	(1) Control (2) Aerobic exercise (3) Resistance exercise (4) Combined training	Not statistically significant in hepatic *Ppar-α* and *Sirt1* gene expression	Exercise improves insulin resistance and decreases ALT levels.	Nikroo, et al., 2020 [30]
Female C57BL/6N mice 3 weeks old n = 44	Aerobic exercise: Running wheels were equipped with tachometers measuring distance (km), average speed (km/h), and time (h:m).	(1) Control sedentary (2) Voluntary wheel running	↑ Hepatic *Ampk*, *Ppar-α* and *Pgc1-α* signaling	Reduction in hepatic lipogenesis and increase in hepatic β-oxidation.	Bae-Gartz, et al., 2020 [31]
C57BL/6 mice n = 6–8/group	Aerobic exercise: Treadmill running 12 m/min, 1 h per day, 5 d/week for 8 weeks.	(1) Control (2) HFD + sedentary (3) HFD + exercise training	Significant inhibition *Ppar-γ* expression by *Il-6*	Prevention of obesity and hepatic fat accumulation.	L. Li, et al., 2021 [32]
Male C57BL/6 mice n = 18	Aerobic exercise: 12–20 m/min for 20–50 min/day at 70% VO_2_ max.	(1) Normal chow diet (2) HFD (3) HFD + exercise	↓ *Fabp4*, *Tgf-β1* and *Col1* mRNA levels ↑ *Ppar-α* Muscle irisin expression	Improvement in lipid metabolism, inhibition of fibrogenesis, and prevention of inflammation.	Zhu, et al., 2021 [33]
Male C57 mice n = 24	Aerobic endurance training: 5 min running at 30–40% of VO_2_ max to warm-up, followed by 30 min running at 60–65% of VO_2_ max and concluded with 5 min cooling down by running at 30–40% of VO_2_ max for 10 weeks.	(1) Normal (2) HFD (3) HFD + exercise	Restoration of the expression of miR-33 ↓ *Srebp1c*, *Fas*, and *Acc* lipogenic gene expression ↑ Autophagy pathway	Beneficial effects on the circulating lipid profile and hepatocyte lipogenesis.	Ghareghani, et al., 2018 [34]
Male C57BL/6J mice 8 weeks old n = 36	Moderate-intensity aerobic exercise: 24 weeks, 10 m/min, 10º inclination, 60 min/day.	(1) Control + sedentary (2) Control + exercise (3) HFD + sedentary (4) HFD + exercise	↑ mRNA expression of *Cbs*, *Ces*, and *3-Mst*, and mitochondrial β-oxidation ↓ *p62*, *Tnf-α* and *Il-6* protein expression	Attenuation of systemic insulin resistance, glucose intolerance, hepatic steatosis, fibrosis, and promotion of autophagy influx in the liver.	Wang, et al., 2017 [35]
Female C57BL/6J mice 9 w old n = 33	Endurance exercise: progressive protocol starting at 12 m/min and ending at 16 m/min for 60 min during 12 w. The intensity was 75−80% VO_2_ max.	(1) Normal-diet control group (2) High-fat diet/high-fructose group (3) HFD/HF + Endurance exercise group	↓ ATP-citrate-lyase and diacylglycerol-O-acyltransferase 1 ↑Oxidative phosphorylation enzymes and acyl-CoA synthetase1	Attenuation of hepatic steatosis, reduction in de novo lipogenesis, enhancement of mitochondrial biogenesis, and fatty-acid activation.	Joshua J. Cook, et al., 2022 [36]
Male C57BL mice 8 weeks old n = 15	Swimming protocol: 5 d/week, 60 min/d for 16 weeks.	(1) Normal diet (2) HFD (3) HFD + Ex	↓ *Srebp1c*, *Scd1*, Fas, *Cd36* and *Acox1* gene expression ↓ MKK4/JNK expression ↑ MIF	Alleviation of hepatic lipid accumulation and reduction in lipotoxicity.	Ni Cui, et al., 2022 [37]
Reduction of lipid biosynthesis					
Male Wistar rats 60 days old n = 40	Strength training: 3 d/week, the animals performed squat jumps in three sets of 12 repetitions with a one-minute pause between sets. The weight load was incremented from 50 to 100% of the animal’s body weight.	(1) Control (2) Strength training (3) HFD (4) HFD + strength training	↓ *Fas/Cd95*, *Limp-II*, *Cd36* and *Srebp-1* proteins.	Reduction in liver glycogen and lipids accumulation.	Dos Santos, et al., 2019 [38]
Male Swiss mice 8 weeks old	Short-term strength training: 20 climbing series with an overload of 70% of the MVCC and with rest intervals of 60–90 s between sets. Total of 13 sessions.	(1) Control lean (2) Obesity	↓ *Fasn* and *Scd1* mRNA levels ↑ *Cpt1a* and *Ppara* mRNA levels, and Akt phosphorylation	Reduction in lipid droplet size, the TG levels in the liver, hepatic lipogenesis, and inflammation	Pereira, et al., 2019 [39]
Male Zucker rats 7 weeks old n = 32	Mixed exercise protocol (interval aerobic training combined with strength): 8 running bouts of 2 min separated by 1 min of rest during which animals ran with a 20° of inclination. The strength exercise was followed by 30 min of aerobic interval exercise, alternating 4 min bouts at 50–65% VO_2_ max with 3 min bouts at a submaximal intensity at 65–85 % VO_2_ max.	(1) Obese (2) lean (3) lean + exercise	↓ the nuclear transcription factor *Srebf1*, *Fasn*, *G6pd* and *Ppara* expression	Reduction in hepatomegaly, steatosis associated with NAFLD, and improvement in glucose and lipid metabolism.	Martínez, et al., 2018 [40]
Male C57BL/6J mice 6 weeks old n = 6/group	Progressive aerobic exercise: Mice swam 5 d/week, 45 min for 8 weeks.	(1) ND sedentary (2) ND + swimming exercise (3) HFD sedentary (4) HFD + swimming exercise	Inhibition of *Hmgcs2* expression and attenuation of the Wnt3a/β-catenin pathway	Prevention of NAFLD-associated liver injury, steatosis, and fibrosis.	Qian, et al., 2021 [41]
Male C57BL/6 mice 4 weeks old n = 34	Aerobic exercise: motorized treadmill at running speeds of 15–20 m/min for 60 min/day, 5 d/week for 16 weeks.	(1) High-fructose water (HFF) sedentary (2) HFF + exercise	↓ *Cd36* and *Ppar-c* expression	Attenuation of hepatic inflammation and fibrosis.	Kawanishi, et al., 2018 [42]
Male C57BL/6J mice 7 weeks old n = 30	Aerobic exercise: swimming training continuously for 30 min/d, 5 d/week for 12 weeks.	(1) Standard diet (2) HFD group with 60% kcal from fat (3) HFD + swimming exercise	↓ *Fabp1* signaling pathway	Alleviation of hepatic steatosis.	Pi, et al., 2019 [43]
Male Sprague-Dawley rats 3 weeks old n = 35	Aerobic exercise: 15 m/min for 30, 45, and 60 min; and 45 min for 20 m/min for 30 or 45 min. Followed by a final 1-week consistent training period, 20 m/min for 60 min. 5 d/week.	(1) HFD + exercise (2) HFD	↓ *Mgat1* pathway	Decrease in hepatic lipid accumulation and improvement in NAFLD.	Baek, et al., 2020 [44]
n = 20 male C57BL/6 J ApoE-KO mice n = 10 Male C57BL/6 J mice	Swim training exercise: 60 min/d, 5 d/week for 12 weeks.	(1) HDF (2) HFD + Ex	↑ PPAR-γ, CPT-1, MCAD expression	Alleviation of lipid metabolism disorders.	Fan Zheng, et al., 2019 [45]
Male C57BL/6 mice 5 weeks old n = 24	Moderate treadmill exercise: running speed increased from 7 m/min to 13 m/min, and duration increased from 15 min to 60 min/d, 5 d/week, for 15 weeks.	(1) Normal diet sedentary (NDS) (2) NDS + Ex (3) HFD (4) HFD + Ex	↓ FITM2, CIDEA, and FSP27	Reduction in hepatic lipid droplet size.	Yangjun Yang, et al., 2022 [46]
Attenuation of the inflammatory response					
Male Sprague Dawley rats 8 weeks old n = 50	Low-intensity exercise training: running at a speed of 15 m/min at a 0° inclination, about 30% VO_2_ max. Moderate-intensity exercise training: running 60 min at a speed of 25 m/min Incremental-intensity progressive exercise: running training intensity was progressively increased from 20 m/min to 30 m/min at 10° inclination for the first 3 weeks. Finally, the last 3 weeks of the exercise was performed in 2 sets of 30 min each, about 75% VO_2_ max. The exercise program involved 5 d/week for 6 weeks.	(1) Normal control (2) HFD (3) Low-intensity exercise (4) Moderate-intensity exercise (5) Incremental-intensity exercise	↓ Endoplasmic reticulum stress signaling pathways IRE1/JNK and eIF2α/CHOP. *↓ Caspase-3*, *Jnk*, *Atf4*, and *Bax* protein expression and hepatocyte apoptosis. ↑ *Bcl-2* protein expression The moderate-intensity exercise demonstrated more effects: ↓ *Ire1*, *eIF2α*, the ratio of *p-Ire1/Ire1*, and *Atf4*	Triglycerides, total cholesterol, fatty free acids, and LDL-c were reduced in all exercise groups. Also, exercise improved lipidemia levels and hepatic injury in NAFLD rats.	Ruan, et al., 2021 [47]
Male C57/BL/6 J mice 7 weeks old n = 16	Aerobic exercise: 50 min, 5 d/week of treadmill exercise for 3 months.	(1) Rest group (2) Exercise group	↓ Inflammatory cytokine (TNF-α and IL-6) levels and induced changes in Kupffer cells capacity.	It may contribute to delaying disease progression in NAFLD.	Komine, et al., 2017 [48]
Male C57BL/6J mice 6 weeks old n = 54	Aerobic exercise: 5 min warmup period at 9 m/min, 50 min main exercise period at 12 m/min (75% VO_2_ max), and a 5 min cooldown period at 9 m/min.	(1) HFD control (2) HFD group (3) HFD + exercise (4) MCD control (5) MCD group, (6) MCD + exercise	↓ NLRP3 inflammasome, Caspase-1 enzymatic activity, and ROS overproduction Normalized *Il-1β* levels	Alleviation of diet-induced hepatic steatosis, inflammation, and fibrosis.	Yang, et al., 2021 [49]
Male C57BL/6J mice 8–10 weeks old n = 15	Aerobic exercise: wheel running activity was recorded using a bicycle tachometer.	(1) Sedentary (2) Voluntary wheel running	↓Macrophage-associated hepatic inflammation.	Improvement in fatty acid and glucose homeostasis.	Gehrke, et al., 2019 [50]
Male Swiss mice 4 weeks old n = 30	Aerobic exercise training: 60% of the EV, 5 d/week, 60 min/day during 8 weeks.	(1) Lean + sedentary. (2) Obese + sedentary. (3) Trained Obese + exercise	↓ CLK2 hepatic content compared to the obese group	Improvement in insulin resistance and attenuation of hepatic fat accumulation.	Muñoz, et al., 2018 [51]
Male Wistar rats 14 weeks old n = 52	Strength exercise protocol: 3 or 4 sets of 2 repetitions each (with 30 s breaks between each repetition and 2 min between each series). Extra weight varied between 0 and 55% of body weight added to the rat’s tails. 3 d/week for 12 weeks.	(1) HFD (2) HFD + supplemental zinc (3) HFD + physical strength exercise (4) HFD with supplemental Zn + strength exercise (ZnEx).	ZnEx ↑ *pAkt* and *p-Ptp1B* levels compared to HFD and Zn groups.	Decrease in IHTG, improvement in insulin signaling, and attenuation of non-alcoholic liver disease.	Vivero, et al., 2021 [52]
Male C57Bl/6 mice 12 weeks old n = 7-10/group	Aerobic exercise: running wheels for 10 weeks.	WT and ERα KO mice: (1) WT sedentary (2) WT wheel running (3) ERα KO sedentary (4) ERα KO + exercise	↓ *Er-α* protein expression	Suppression of hepatic steatosis and inflammatory gene transcripts in WT but not KO mice.	Winn, et al., 2019 [53]
Male Wistar rats 9 weeks old n = 16	Endurance exercise: treadmill running protocol 5d/week, progressively increasing the time from 30 min to 1 h every week and speed from 10 to 19 m/min weekly for 7 weeks.	(1) Sedentary (2) Endurance exercise	Failed to improve the expression of complexes I and II of the respiratory chain and *Irs2*	Exercise did not improve the NASH activity score; however, it reduced hepatic cholesterol.	Henkel, et al., 2019 [54]
Male C57/BL6J mice 30 weeks old Male p62-KO mice 30 weeks old	Aerobic exercise: protocol of 10, 12, 14, and 16 m/min for 5 min each and at 18 m/min for 30 min (50 min total), 5 d/week for 4 weeks.	(1) WT group, (2) 1-month resting *p62*-KO group (*p62*-KO-Rest) (3) 1-month exercising *p62*-KO group (*p62*-KO-Ex),	Impairment of phagocytic capacity of KCs through greater DHEA production	Decrease in hepatomegaly, hepatic inflammation, and fibrosis.	Miura, et al., 2021 [55]
Male Sprague Dawley rats 8 weeks old n = 24	Swim training exercise: for 5 d/week, 60 min/d, for 6 weeks.	(1) STD (2) STD + exercise (3) HFD (4) HFD + exercise	Amelioration of oxidative change: MDA GSH levels	Decrease in steatosis, hepatocellular ballooning, inflammation, fibrosis, and glycogen content.	Açıkel Elmas, et al., 2020 [56]
Male C57BL/6 mice 8 weeks old n = 40	Aerobic exercise: 12 m/min, 60 min/day, 5 d/week, for 8 weeks.	(1) Control group (2) Control + exercise group (3) NAFLD model group (4) NAFLD model + exercise group	↓ CNPY2-PERK pathway	Improvement in liver histology.	Junhan Li, et al., 2022 [57]
Male C57L/6 mice 5 weeks old n = 30	Voluntary wheel running for 12 weeks.	(1)Control group (2) High-fat diet sedentary group (3) High-fat diet + voluntary wheel running group	↑ gluconeogenesis, detoxification, mitochondrial biogenesis, and proteolysis pathways in the liver.	Improvement in pathophysiological conditions.	Byunghun So, et al., 2022 [58]
Male C57BL/6 mice 3 month old n = 80	High-intensity interval training: animals performed five bouts (2 min) at 80% (w1)–85% (w2)–90% (w3) and 95% (w4), with five active rest periods of 1 min between bouts at 50% MRC, 3 d/week for 4 weeks.	(1) Control (2) Control + IF (3) Control + HIIT (4) Control + IF/HIIT (5) High-Fat (HF) diet (6) HF + IF (7) HF + HIIT (8) HF + IF/HIIT	↓ IL-6, MCP-1, and PAI-1.	Improvement in glucose tolerance/insulin resistance, liver steatosis/inflammation, fatty acid oxidation, and lipogenesis.	Patrícia de Castro-de-Paiva, et al., 2022 [59]
Male Sprague–Dawley rats 5–6 weeks old n = 36	Aerobic exercise: 5 d/week, 60 min/day, at a starting speed of 15 m/min, which was gradually increased over the training program up to 25 m/min.	(1) SCLD + sedentary animals (2) SCLD + voluntarily physically active animals (3) SCLD + endurance-trained animals (4) liquid HFD + sedentary animals (5) liquid HFD + voluntarily physically active animals (6) liquid HFD + endurance-trained animals	Physical exercise counteracts NASH-related ER stress	Modulates non-alcoholic steatohepatitis-related hepatic endoplasmic reticulum stress.	Emanuel Passos, et al., 2022 [60]
Male C57BL/6J mice	Treadmill training for 8 weeks.	(1) knocking down for Higd1a (2) Overexpressing Higd1 mice	↑ Higd1a	Alleviation of hepatic steatosis, liver injury, and inflammation.	Jie-Ying Zhu, et al., 2022 [61]
Sprague–Dawley rats n = 24	Swim training exercises: for 3 d/week, 60 min/d, for 4 weeks	(1) Control (2) Control + Ex (3) High-cholesterol and fructose diet (HCFD) (4) HCFD + Ex	↓ *Tnf-α*, *il-6*	Improvement in steatosis, hepatic enzymes, lipid profile, glucose homeostasis, inflammatory biomarkers, and substantial restoration of the normal hepatocyte ultrastructure.	Mohammed A. Dallak, et al., 2018 [62]
Female Sprague- Dawley rats 8–10 weeks old n = 24-36	Aerobic exercise protocol: 30 min/d, 6 d/week, for 9–12 weeks.	(1) normal diet (2) normal diet + Ex (3) HFD (4) HFD + Ex	↓ *Tnf-α*, *il-6*, *NF-κB*	Decrease in inflammation levels and improvement in muscle insulin sensitivity.	Qian Yu, et al., 2019 [63]
Male C57BL/6 Mice 8 weeks old	Endurance exercise training program: 12 to 19 m/min (increasing from 30 to 70 min day), 5 d/week, for 6 weeks.	(1) Control HFD (2) HFD + Ex (3) Sdc4 AAV administration	↑ Sdc4 Proteomic changes in hepatocytes	Reduction in fatty acid uptake, macrosteatosis, and expression of inflammatory and profibrotic genes in the liver.	William De Nardo, et al., 2022 [64]

Abbreviations: 3-MST: 3-mercaptopyruvate sulfurtransferase; AAV: adeno-associated virus; ACOX1: fatty acid oxidation genes; Ampk: AMP- activated protein kinase; Atf4: activating transcription factor 4; Atgl: adipose triglyceride lipase; Cbs: cystathionine β-synthase; Ces: cystathionine γ-lyase; CHOP: C/EBP homologous protein; Cpt-1: carnitine palmitoyltransferase-1; CptIA: carnitine palmitoyltransferase 1A; Dhea: dehydroepiandrosterone; eIF2α: eukaryotic translation initiation factor 2α; Fabp1: fatty acid-binding protein; Fas: fatty acid synthase; Gsh: glutathione; Het: liquid HFD + endurance-trained animals; HFD: high-fat diet; HFHFD: high-fat high-fructose diet; HIIT: high-intensity interval training; Hmgcs2: 3-hydroxy-3-methylglutaryl-CoA synthase 2; HS: liquid HFD + sedentary animals; HVPA: liquid HFD + voluntarily physically active animals; IF: intermittent fasting; IRE1: inositol-requiring enzyme 1α; IRS2: insulin receptor substrate 2; JNK: Jun N-terminal kinases; KCs: Kupffer cells; LIF: leukemia inhibitory factor; MCD: methionine and choline-deficient; MDA: malondialdehyde; Mgat1: monoacylglycerol O-acyltransferase 1; MRC: maximal running capacity; MVCC: maximal voluntary carrying capacity; NLRP3: NACHT, LRR, and PYD domains containing protein 3; p-Ampk: phosphor-AMP-activated protein kinase; Pgc1α: PPAR coactivator-1 alpha; Ppar-γ: peroxisome proliferator-activated receptor; Ppar-α: peroxisome proliferator-activated receptor alpha; Pppar-α, Ppar-δ and Ppar-γ: peroxisome proliferator-activated receptor α, δ and γ; Sdc4: Syndecan-4; SCLD: standard control liquid diet; Scd1: stearoyl CoA desaturase-1; SET: standard liquid diet + endurance-trained animals; ER: endoplasmic reticulum, Sirt1: Sirtuin-1; Sra: steroid receptor RNA activator; Srebf1 and Srebf2: sterol regulatory element binding transcription factor 1 and 2; SS: standard liquid diet + sedentary animals; STO: strength training obese; SVPA: standard liquid diet + voluntarily physically active animals; Tgf-β: transforming growth factor-β; VPA: voluntary physical activity. ↑ upregulation and ↓ downregulation.

### 3.1. Experimental Model

#### 3.1.1. Exercise Reduces Lipid Accumulation in the Liver

The pathogenesis of NAFLD involves lipid accumulation, inflammation, fibrosis, and disruption of liver homeostasis. Nevertheless, lifestyle changes may reduce the damage and even reverse the common lesions of NAFLD [22,65]. Physical activity modifies the cellular and molecular pathways in patients with hepatic steatosis, though the specific responses remain unclear.

A large number of experimental model studies suggest that exercise intervention may reduce the pathological markers of NAFLD, such as lipid droplet size, fibrosis, vesicular steatosis, hepatocellular ballooning, as well as triglycerides, cholesterol, and glycogen content. In addition, physical activity improves insulin signaling, glucose tolerance, LDL-c, and hepatic β-oxidation [23,24,25,26,27,38,47,66]. The best available evidence of exercise-treated mice showed amendment of NAFLD onset and change in lipogenic gene expression. An aerobic exercise protocol increased the phosphorylation of AMP-activated protein kinase (AMPK), which plays a major role in the regulation of cellular energy balance and increase in the rate of catabolic pathways [67]. Evidence has shown that increase in lipogenic factors, starting with AMPK, could upregulate hepatic carnitine palmitoyl-CoA transferase 1 (CPT1) and adipose triglyceride lipase (ATGL) [23,26,27]. CPT1 modulates the mitochondrial fatty acid β-oxidation pathway [68]. Accordingly, OK et al., Li et al., and Melo et al. have reported an increase in CPT1 in response to aerobic exercise [26,28,29], while Pereira et al. and Bae et al. also reported this after short-term strength training and ladder climbing exercise, respectively [23,66]. The main results were reduction in the size of liver fat droplets and reversal of high-fat diet (HFD)-induced steatosis.

Lipid accumulation is also associated with peroxisomes proliferator-activated receptors (PPARs) activity. PPARs are intimately related to hepatic lipid metabolism regulation [30]. Positive changes were observed in *Ppar-α*, *Ppar-δ*, *Ppar-γ*, and PPAR coactivator-1 alpha (*Pgc1-α*) after aerobic exercise training [22,25,29,31,32,33]. Also, experimental models have shown reductions in the size of liver fat droplets, inhibition of hepatic lipogenesis, reversal of HFD-induced steatosis, and decrease in inflammation. Other authors have also reported upregulation of *Sirt1* signaling [25,28] and *Cyp4A10* signaling [29] molecules involved in mitochondrial β-oxidation and autophagy markers after a moderate aerobic training protocol [28,34,35]. However, some controversy remains, since Nikroo et al. could not replicate the increase in hepatic *Ppar-α* and *Sirt1* expression after aerobic training [30].

#### 3.1.2. Exercise Reduces Lipid Biosynthesis in the Liver

The accumulation of lipids in the liver is directly linked to hepatic lipogenesis. Several studies have shown the potential effectiveness of exercise programs, mostly aerobic training, in reducing hepatic steatosis by upregulating β-oxidation. Therefore, exercise training may also downregulate lipid synthesis in the liver [24,38].

Fatty acid synthase protein (FAS) plays a role in the synthesis of palmitic acid, while stearoyl CoA desaturase-1 (SCD1) is involved in the formation of monounsaturated fatty acids. Both enzymes catalyze rate-limiting steps in the pathways for fatty acid synthesis. Since the expression of Fasn and Scd1 is increased in mice with fatty liver, they have been proposed as key proteins in NAFLD development. Although the literature provides more evidence about the beneficial effects of aerobic training, short-term strength training also decreases Fasn and Scd1 [39]. Similarly, high-intensity interval training [24], strength training [38], and a mixed exercise protocol [40] have been found to decrease Fas in obese experimental models. Furthermore, other key proteins involved in cholesterol synthesis, such as sterol regulatory element-binding proteins 1 and 2 (SREBP-1/2) and 3-hydroxy-3-methylglutaryl-CoA synthase 2 (HMGCS2), are decreased in response to physical activity regardless of the type, intensity, and intervals of exercise [23,24,29,34,40,41].

The scavenger receptor CD36 functions as a central signaling protein in lipid metabolism and has been observed to increase with the use of HFD [43]. In this context, both strength and endurance training have been shown to reduce the obesity-related increase in CD36 [29,38,42]. Fatty-acid-binding protein (FABP) is another protein closely associated with lipid metabolism, which was also downregulated in trained NAFLD mice [33,43].

Other molecules involved in hepatic lipid synthesis have been measured after several exercise protocols. Despite the differences in the protocols, they resulted in a general downregulation in biosynthesis genes. Particularly, high-intensity interval training (HIIT) showed a better effect on the genes related to lipogenesis, such as Pparg, diacylglycerol O-acyltransferase 1 (Dgat1), acetyl-CoA carboxylase alpha (Acaca) and acetyl-CoA carboxylase beta (Acacb), than moderate-intensity continuous training (MIT) [22]; acetyl-CoA carboxylase (Acc) was also reduced when compared to endurance training [24]. However, other authors also showed a decrease in Acc lipogenic gene expression after aerobic endurance training [34], as well as monoacylglycerol O-acyltransferase 1 (Mgat1), which plays a role in the synthesis of diacylglycerol (DAG) and triacylglycerol (TAG) [44].

#### 3.1.3. Exercise Attenuates the Inflammatory Response

The obesity-related accumulation of adipose tissue leads to lipolysis and higher transfer of lipids to different tissues, hence increasing de novo hepatic lipogenesis and decreasing free fatty acid oxidation [69].

In addition, hepatocytes also suffer from high lipid deposition by the transferring of free fatty acids from the adipose tissue, which is usually associated with an inflammatory state [27,47]. Furthermore, the deposition of free fatty acids in the mitochondria may induce the production of reactive oxygen species (ROS), tumor necrosis factor-alpha (TNF-α), as well as other pro-inflammatory cytokines associated with hepatic endoplasmic reticulum (ER) stress [69,70].

NAFLD provokes fibrosis in the context of inflammation-mediated ER stress, proapoptotic cascades, and apoptosis of hepatocytes. Additionally, hepatic ER stress is a major stimulus that induces the recruitment of immune cells to the damaged tissue, which may ultimately lead to terminal organ failure derived from the exacerbated production of proinflammatory cytokines [65,70].

Several experimental studies have shown that exercise intervention, as a first-line treatment for NAFLD, confers significant hepatic protection. Moderate-intensity aerobic exercise differentially modified the inflammatory genes expression *p62*, *Tnf-α*, and *Il-6* [35]. Moreover, aerobic training also suppressed the inflammation markers TNF-α and IL-6 [48]. It also suppressed the steroid receptor RNA activator (SRA)/JNK/P38 signaling pathway, leading to improved hepatic steatosis and a decrease in the production of inflammatory cytokines [29]. Ruan et al. studied the impact of different levels of exercise intensity and showed that moderate-intensity exercise inhibited hepatocyte apoptosis through a signaling pathway associated with the ER: inositol-requiring enzyme 1α (IRE1)/Jun N-terminal kinases (JNK) and eukaryotic translation initiation factor 2α (eIF2α)/C/EBP homologous protein (CHOP) [47]. Also, Lou et al. investigated the combined effect of aerobic and resistance exercise in GCN2KO mice and showed beneficial effects towards reversing hepatic steatosis by downregulation of *eIF2α* and activation of transcription factor 4 (*Atf4*) [25].

NAFLD progression is associated with increased levels of ROS as a marker of poor prognosis. ROS stimulate and promote inflammasome activation through the NACHT, LRR, and PYD domains containing protein 3 (NLRP3), which is highly expressed in the liver and is best characterized by its close association with several chronic diseases [71]. The NLRP3 inflammasome also increases the secretion of the proinflammatory cytokines IL-1β and IL-18 [72]. Aerobic exercise protocols significantly reduce the *Nlrp3* multiprotein complex, normalize *Il-1β* production, and suppress ROS overproduction [49]. Additionally, Gherk et al. showed that aerobic physical activity protects the tissue from macrophage-associated hepatic inflammation in an NAFLD mouse model [50]. Figure 2 shows the effect of exercise training on metabolic pathways in a fatty liver animal model.

### 3.2. Human Model

The results of the quality assessment for the RCT included are displayed in Figure 3. Three studies showed some concerns for risk of bias in their randomization process because they did not provide details of the random allocation sequence. Since physical exercise was the primary intervention among all the RCT included, they did not adopt a blind process and participants were aware of their assigned intervention during the trail. Therefore, all the studies were judged with concerns over possible deviations from the intended interventions and that the assessment of the outcome could be compromised by the knowledge of the intervention received by evaluators and participants. In this way, measures of the outcome were judged with high risk of bias in all the studies. However, no other domains were judged to have high risk of bias.

Physical activity is defined as any movement of the body produced by skeletal muscles that requires more energy than is consumed while resting. In addition, exercise is a subcategory of physical activity defined as planned, structured, and repetitive movements to maintain or improve fitness [73]. Interventions involving the introduction of exercise in lifestyle are an effective strategy to modify the metabolism of hepatic fatty acids and reduce total triglycerides with or without weight loss [74,75].

Exercise is usually classified as aerobic or resistance exercise. Aerobic exercise is generally known as “cardio”, which strengthens heart and lung capacity and improves the consumption of oxygen in the body. Aerobic exercise develops mainly type I muscle fibers by increasing their aerobic capacity, while also improving the cardiorespiratory system through the enhancement of microcirculation and arterial compliance, as well as strengthening the respiratory muscles and enhancing myocardial contractility [76]. In addition, it activates lipolysis in several tissues, upregulates the uncoupling of protein-1 and peroxisome proliferator-activated receptor γ pathways, and modifies adipokine levels [7,77].

Resistance exercise, on the other hand, builds up muscle strength and improves muscle tone and bulk [78,79]. This type of exercise promotes the hypertrophy of type II muscle fibers and aims to increase myokine levels, activate glucose transporter 4, caveolins, and the AMP-activated protein pathway, as well as to improve NAFLD through less energy consumption during exercise [77].

Standardizing physical activity prescriptions for NAFLD patients remain a challenge. The EASL-EASO-EASD guidelines for the management of NAFLD recommend 150–200 min per week of moderate-intensity aerobic physical activities in 3–5 sessions while emphasizing the efficacy of resistance training in reducing liver fat and improving musculoskeletal fitness and metabolic risk factors. The guidelines also emphasize the importance of tailoring the training approach to individual patient preferences [18]. The American Association for the Study of Liver Diseases (AASLD) recommends ≥ 150 min/week of moderate-intensity exercise [17], similar to the AGA advice to target 150–300 min of moderate-intensity or 75–150 min of vigorous-intensity aerobic exercise, with consideration of resistance training only as a complement [8]. Moreover, other guidelines only make non-specific recommendations to increase physical activity [80], attending to patient preferences for aerobic or resistance training to ensure long-term adherence, considering that resistance exercise may be more feasible than aerobic exercise in patients with poor fitness [81].

Although lifestyle interventions, such as changes in diet and exercise to achieve weight loss, remain the cornerstone of NAFLD treatment due to their additive effects in reducing liver fat content, the benefits of each of them have been widely demonstrated independently [17,18,82,83,84]. Several randomized clinical trials have shown that exercise alone improves NAFLD even without dietary restrictions [85,86,87,88,89,90,91].

#### 3.2.1. Weight Loss in NAFLD

Weight loss is considered a cornerstone to resolve steatosis, NASH, and liver fibrosis [92]; however, exercise can improve NAFLD even without achieving weight loss. Houghton et al. found that 12 weeks of exercise in the absence of weight loss reduced 16% liver fat, 12% visceral fat, and 23% serum triglycerides in patients with biopsy-proven NASH [86]. These results are in line with other studies that demonstrated that an exercise intervention program conferred significant improvements in hepatic stiffness and fat content [85,87,93], glucose homeostasis, and lipid metabolism, regardless of weight loss [91].

**Figure 3 healthcare-11-01992-f003:**
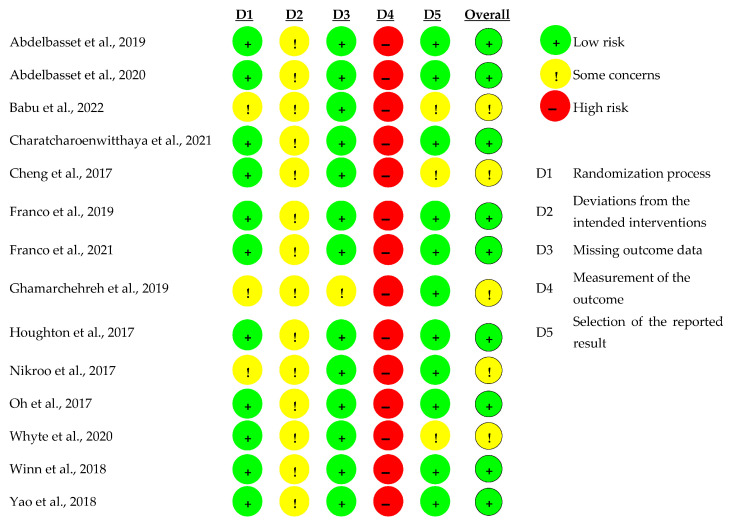
Quality assessment of the RCT included [82,83,84,85,86,87,88,89,90,91,93,94,95,96].

#### 3.2.2. Aerobic Exercise vs. Resistance Exercise

Both aerobic and resistance exercises have shown benefits in patients with NAFLD [82,83,86,90], which have led to several RCTs to investigate which one provides greater benefits. Yao et al., evaluated the effects of aerobic and resistance exercise on ALT and blood lipids in patients with NAFLD and they observed significant improvement in HDL in both groups, but only aerobic exercise improved serum ALT and triglycerides after 22 weeks of training [94]. Ghamarchehreh et al. found that aerobic training was more effective than resistance training for improving the lipid profile, particularly total cholesterol, HDL, and LDL in elderly subjects with NAFLD, in an 8-week aerobic and resistance training program [95]. However, Charatcharoenwitthaya et al. found no differences in the reduction in hepatic fat content, abdominal adiposity, or improvement in insulin resistance after 12 weeks of aerobic or resistance training in patients with NAFLD [93].

#### 3.2.3. Aerobic vs. Combined Aerobic plus Resistance Exercise

The response of patients with NAFLD to either aerobic exercise or a combination of aerobic plus resistance exercise was evaluated by Franco et al. They found that after 6 months of exercise training, both groups significantly reduced their NAFLD mean score, but the aerobic exercise program was more effective [89]. Subsequently, Franco et al. evaluated the effects of an aerobic exercise program, an aerobic plus resistance program, a low-glycemic index Mediterranean diet (LGIMD), and their combined effects, in patients with NAFLD, finding that after 90 days, all interventions significantly reduced the NAFLD score, but the LGIMD plus aerobic activity program was associated with the stronger reduction [84].

#### 3.2.4. Moderate-Intensity Continuous Aerobic Training vs. High-Intensity Interval Training

The relative intensity of an exercise is defined as the percentage of utilized aerobic power and is expressed as the percentage of maximum heart rate or percentage of VO_2_ max. It is important to consider the relative intensity since it is an important factor in the management of NAFLD [73]. Accordingly, physical activities classified as moderate-intensity are performed at a relative intensity of 40–60% VO_2_ max, while vigorous-intensity activities are performed at a relative intensity of >60% VO_2_ max, in line with high-intensity exercise [73].

Several studies have evaluated the impact of the intensity of aerobic exercise on the management of NAFLD. Oh et al., compared the therapeutic effects of resistance training and two intensity modalities of aerobic training, high-intensity interval training (HIIT) and moderate-intensity continuous aerobic training (MICT), for 12 weeks in sedentary obese subjects with NAFLD. All three exercise modalities were equally effective in reducing hepatic fat content, regardless of significant weight and visceral reductions, but only the HIIT group significantly improved their hepatic stiffness associated with restoring the Kupffer cell function and decreased their inflammation markers (leptin and ferritin) [85]. Babu et al. found a significant decrease in blood glucose and waist circumference, as well as an increase in VO_2_ max with a 12-week HIIT exercise program compared to a non-intervention group in subjects with NAFLD. In addition, the exercise program group benefited from the loss of weight and promoted positive metabolic changes in amino acids, lipids, and bile acids involved in the regulation of glucose metabolism [91].

Abdelbasset et al. found significant beneficial effects after an 8-week HIIT intervention on intrahepatic triglycerides (IHTG), visceral lipids, and health-related quality of life in diabetic obese individuals with NAFLD [88]. They also assessed the effects of HIIT and MICT for 8 weeks in diabetic obese patients with NAFLD and found that both training modalities significantly reduced IHTG, visceral lipids, plasma ALT, plasma glucose, and improved insulin sensitivity, even though no differences were observed between both exercise modalities [96]. These results are consistent with Winn et al., who demonstrated a significant reduction in intrahepatic lipid (IHL) content after a 4-week training program with either HIIT or MICT matched for energy expenditure (~400 kcal/session) in patients with NAFLD. Nevertheless, the differences in IHL reduction were not significant between both exercise intensities (*p* = 0.25) [87].

#### 3.2.5. Exercise Dose in NAFLD

The optimal dose of exercise prescription to achieve results for the proper management of NAFLD is not clear since differences exist regarding the amount of exercise per session, times per week, intensity, and the total amount of exercise to observe an improvement in subjects with NAFLD. One study found a significant correlation between the number of sessions per week and the absolute reduction in hepatic fat content (r = 0.52; *p* = 0.001) [93]. Programs of three sessions/week have shown optimal results regarding NAFLD resolution marked by an improvement in insulin sensitivity and a reduction in fasting plasma glucose in several RCTs with both aerobic and resistance exercise [82,84,85,88,94,95,96].

There are differences in the duration of the intervention in training programs regarding their effectiveness in the management of NAFLD. Winn showed a significant reduction in intrahepatic lipid content with just 4 weeks of aerobic training in subjects with NAFLD [87]; however, most studies have evaluated 8–12 weeks of exercise to observe the effectiveness of the intervention to improve or resolve NAFLD [82,83,86,88,93,95,96]. Some studies have even extended up to 22 weeks [94] or 6 months of intervention and have achieved higher improvements in the metabolic parameters and the NAFLD score [89].

#### 3.2.6. Optimal Exercise Modalities in NAFLD

Although various exercise modalities seem to improve or even resolve NAFLD, their prescription must be considered within the context of several aspects related to the patient, such as age, comorbidities, fitness status, and preferences [76], since aerobic and resistance exercise have different advantages and disadvantages. There exists more evidence of effectiveness for aerobic exercise than resistance exercise regarding the improvement in visceral adipose tissue, ALT, glucose, and lipids [94,95,97]. However, it requires higher cardiorespiratory fitness because it causes fatigue and less tolerance (particularly HIIT), which may lead to poor compliance. Moreover, patients with comorbidities, such as coronary disease and osteoarthritis (frequently in the knees), could limit their ability, even though moderate-intensity activities, such as brisk walking, jogging, and cycling, are simple and cost-effective [7,76].

On the other hand, resistance exercise requires less energy consumption and may be better tolerated in patients with poor cardiorespiratory fitness. On the downside, it frequently requires specialized machines and equipment coupled with a personal trainer, at least initially, to perform the exercises properly and reduce the risk of musculoskeletal injuries that may eventually compromise exercise compliance [76].

#### 3.2.7. Exercise Adherence in NAFLD

Adherence to any type of exercise represents an issue for many people. In a study of patients with NAFLD, Stine et al. found 75% of patients failed to achieve the prescription of ≥150 min/week of physical activity. Lack of resources and education from their treating medical provider, physical discomfort, and time restrictions were the major barriers identified by patients [98]. Therefore, the modality of exercise, the number of sessions per week, and the intensity must be individualized according to the age, fitness, comorbidities, and preferences of each patient, seeking greater compliance in the long term by finding the most tolerable and enjoyable activities for each patient.

**Table 2 healthcare-11-01992-t002:** The effects of physical activity in randomized controlled studies with supervised training in patients with NAFLD.

Population	Intervention/Allocation (n)	Exercise					Outcomes	Reference
		Type of Activity	Intensity	Frequency (t/week)	Duration (min)	Intervention Duration (weeks)		
24 subjects with NASH	Exercise: 1. Aerobic + resistance (12) 2. Control (12)	Aerobic + resistance	Borg rating [99] 16 to 18 (very hard)	3	45–60	12	Exercise significantly reduced HTGC, visceral fat, TG, and GGT regardless of weight loss.	Houghton et al., 2017 [86]
63 pre-diabetic patients with NAFLD	1. Exercise (29) 2. Diet (28) 37–40% CHO, 9–13 g fiber, 35–37% fat, 25–27% protein 3. Exercise + diet (29) 4. No intervention (29)	Aerobic	60–65% VO_2_ max	2–3	30–60	8.6 months	HFC was significantly reduced in all the groups compared with the non-intervention group. Only the exercise + diet (fiber-enriched) group significantly decreased their HbA1c levels.	Cheng et al., 2017 [83]
25 men with NASH	1. CR diet + E (12) 2. CR diet (13). CR diet: 500 kcal less than required, 60% CHO, 25% fat, 15% protein	Aerobic: walking, jogging, or running	55–60% HRR	3	35–50	8	Significant improvement in BP, FPG, TG, HOMA-IR, liver steatosis, and QoL only in the CR diet + E group. WC, WHR, ALT, and VO_2_ peak improved in both groups, but improvement was significantly higher in the CR-diet + E group.	Nikroo et al., 2017 [82]
52 sedentary obese men with NAFLD	Exercise: 1. RT (19) 2. HIAT (20) 3. MICT (13)	Resistance or aerobic: HIAT, MICT	HIAT = 80–85% VO_2_ max MICT = 60–65% VO_2_ max	3	HIAT = cycling for 3 min sessions MICT = cycling for 40 min	12	Hepatic fat content was significantly reduced in all 3 groups (RT, HIAT, and MICT), regardless of reductions in weight and visceral fat. Significant improvement in hepatic stiffness only in the HIAT group, which was associated with restoration of Kupffer cell function and decrease in inflammation markers (leptin and ferritin).	Oh et al., 2017 [85]
16 obese with NAFLD	Exercise: 1. HIIT (8) 2. MICT (8) 3. Obese subjects (control) (5)	Aerobic: HIIT or MICT. Energy expenditure = ~400 kcal/session	HIIT = 80 VO_2_ peak/50% VO_2_ peak MICT = 55% VO_2_ peak	4	HIIT= 4 min/3 min recovery MICT= 60 min	4	HIIT and MICT significantly reduced IHL without significant differences between groups. No significant changes in body mass, abdominal adiposity, or biomarkers of liver function were observed between groups.	Winn et al., 2018 [87]
91 subjects with NAFLD	Exercise: 1. Aerobic (29) 2. Resistance (31) 3. Control (31)	Aerobic or resistance	60–70% max HR	3	60	22	A significant improvement in HDL levels in both groups. Additionally, the aerobic exercise group significantly reduced serum ALT and TG.	Yao et al., 2018 [94]
32 diabetic obese subjects with NAFLD	Exercise: 1. HIIT (16) 2. Control (16)	Aerobic: Moderate-to-vigorous intensity	80–85% VO_2_ max /Interval 50% VO_2_ max	3	Total = 40 HIIT = 4 min/2 min interval	8	VO_2_ peak, BMI, IHTG, VAT, plasma lipids, ALT, HbA1c, HOMA-IR, and HRQoL were significantly reduced in the HIIT group.	Abdelbasset et al., 2019 [88]
94 patients with moderate or severe NAFLD	Exercise: 1. Aerobic (42) 2. Aerobic + resistance (52)	Aerobic or aerobic+ resistance	65–75% VO_2_ max	4	Aerobic:45 Combined:70 (30 min aerobic + 40 min resistance)	6 months	NAFLD mean score was significantly reduced in both groups. However, the reduction was more significant in the aerobic exercise group than in the combined exercise group.	Franco, et al., 2019 [89]
39 elderly patients with NAFLD	Exercise: 1. Aerobic (13) 2. Resistance (13) 3. Control (13)	Aerobic or resistance	Aerobic: 55–75% HRR Resistance: 50–70%	3	45 min	8	A significant decrease in the levels of cholesterol and LDL and an increase in HDL was observed in the aerobic group compared with the resistance and control groups.	Ghamarchehreh et al., 2019 [95]
27 men with NAFLD	Exercise: 1. Aerobic (15) 2. Control (12)	Aerobic	40–60%	4–5	Progressive 20 to 60 min	16	Exercise significantly reduced BMI, HFC, FBG, HOMA 2, ALT, and LDL, and also increased VO_2_ Max. No significant changes were observed in HDL kinetics.	Whyte et al., 2020 [90]
47 diabetic obese individuals with NAFLD	Exercise: 1. HII (16) 2. MIC (15) 3. Control (16)	Aerobic: HII or MIC	HII:80–85% VO_2_ max/Interval 50% VO_2_ max MIC: 60–70% max HR	3	HII = 40 min HII = 4 min/2 min interval MIC = 40–50 min	8	Both HII and MIC exercise programs significantly reduced BMI, IHTG, visceral lipids, insulin resistance, ALT, and HbA1c. No differences were observed between both groups.	Abdelbasset et al., 2020 [96]
35 subjects with NFLD	Exercise: 1. Aerobic (18) 2. Resistance (17)	Aerobic or resistance	60–70/ max HR	5	60 min	12	A significant reduction in HFC, hepatic steatosis, WC, and HOMA-IR were observed in both groups, regardless of weight loss. A significant correlation was observed, where more exercise sessions/week correlated with the HFC reduction.	Charatcharoenwitthaya et al., 2021 [93]
144 subjects with moderate or severe NAFLD	1. CD (22) 2. LGIMD (23) 3. PA1 (25) 4. PA2 (23) 5. LGIMD + PA1 (27) 6. LGIMD + PA2 (24)	PA1 = aerobic PA2 = aerobic + resistance	60–75% max HR	3	PA1 = 50–60 PA2 = 60–80	90 days	Significant reduction in NAFLD score in all the groups vs. CD. The best results were obtained in the LGIMD + PA groups, mainly in the PA1 (aerobic) + LGIMD.	Franco et al., 2021 [84]
46 subjects with NAFLD	Exercise: 1. Aerobic, HIIT (21) 2. Control (25)	Aerobic HIIT	85% of maxW4	2	40–50	12	HIIT significantly decreased FPG, and WC, and increased VO_2_ max without weight loss. It also promoted metabolic changes in amino acids, lipids, and bile acids involved in the regulation of glucose metabolism.	Babu AF et al., 2022 [91]

Abbreviations: NASH: non-alcoholic steatohepatitis; NAFLD: non-alcoholic fatty liver disease; CR: calorie-restricted; CHO: carbohydrates; E: exercise; HR: heart ratio, HRR: heart rate reserve; BP: blood pressure; FG: fasting glucose; TG: triglycerides; HOMA-IR: homeostasis model assessment of insulin resistance; HRQoL: heart-related quality of life; WC: waist circumference; WHR: waist-to-height ratio; VO_2_ peak, peak oxygen consumption; RT: resistance training; HIAT: high-intensity interval aerobic training; MICT: moderate-intensity continuous training; NA: not applicable; HIIT: high-intensity interval training; IHL: intrahepatic lipid; CD: controlled diet; LGIMD: low glycemic index Mediterranean diet; PA1: physical activity 1; PA2: physical activity 2.

## 4. Strengths and Limitations

Our review was carefully revised to ensure that only RCTs were included, which deliver a high level of evidence, thus preventing selection bias. We assessed whether the exercise training was performed under supervision, ensuring correct compliance to the intervention. We only included studies that provided detailed information related to the exercise training modality, intensity, frequency, session duration, and intervention period. However, our review still has several limitations. This study focused primarily on the effects of exercise training, while the impact of studies on the effects of diet on NAFLD was seldom considered and, thus, the effect of diet coupled to exercise could not be evaluated; additionally, we excluded cohort studies. The heterogeneity of the studies regarding comorbidities of the study population, NAFLD stages, assessment of exercise modalities, intervention period, as well as ethnicity and outcome assessment, represents another limitation. The included studies measured different aspects, so that, although most of them evaluated changes in hepatic fat content with exercise training, others only evaluated biochemical parameters. Since only studies on adults were included, these results cannot be generalized to pediatric or adolescent populations. Finally, only studies published in English were included in the present review.

The intervention for most studies ranged between 8 and 12 weeks, and, therefore, the effects of the exercise training in the long term could not be determined. Furthermore, no study evaluated long-term outcomes after the intervention had ended. Therefore, future research should focus on evaluation of the effect of post-intervention exercise in participants who had a resolution of NAFLD with exercise training, and include comparison of studies that include combined intervention with those with exercise alone to evaluate the potential of exercise. Additionally, more variables should be measured, such as the ethnicity of the population, lifestyle habits, and pathological history, that can guide us to generate increasingly specific exercise strategies for NAFLD patients.

## 5. Conclusions

The implementation of physical activity showed a strong association with improvements in inflammation, steatohepatitis, and fibrosis, and a beneficial effect on liver function in experimental models. In addition, physical activity demonstrated other major benefits, e.g., the suppression of genes related to lipogenesis and inflammation, as well as upregulation of those related to lipid oxidation and the apoptosis pathway in the liver. Several exercise modalities were demonstrated to have a positive effect in clinical studies of NAFLD in humans. An optimal exercise prescription in terms of type, intensity, and dose that improves or resolves NAFLD has not been established; nevertheless, a dose-response relationship has been observed. Both aerobic exercise and resistance exercise have been demonstrated to reduce liver fat and improve insulin resistance, and blood lipids regardless of weight loss, although there is more evidence of positive effects for aerobic exercise. Resistance exercise is more feasible for NAFLD patients with poor cardiorespiratory fitness. Short-term training programs have proved to be effective, but the benefits may be lost in the long term without proper adherence to permanent lifestyle modification. Diet and exercise prescriptions for NAFLD should be individualized according to the preference, physical fitness, and comorbidities of each patient to promote sustained adherence to lifestyle changes. More effort and awareness-raising should be applied to encouraging an active lifestyle for a better impact on NAFLD patients, and, therefore, reduction in the burden associated with this growing public health problem.

## Figures and Tables

**Figure 1 healthcare-11-01992-f001:**
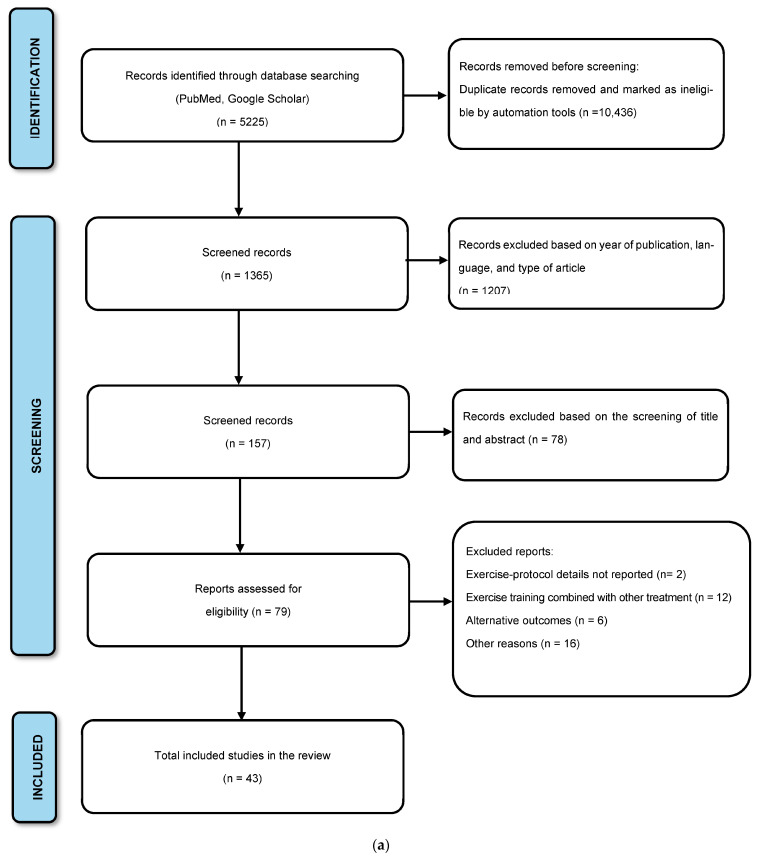
(**a**) Preferred Reporting Items for Systematic Reviews diagram shows the study selection process for experimental models. (**b**) Preferred Reporting Items for Systematic Reviews diagram shows the study selection process for human studies.

**Figure 2 healthcare-11-01992-f002:**
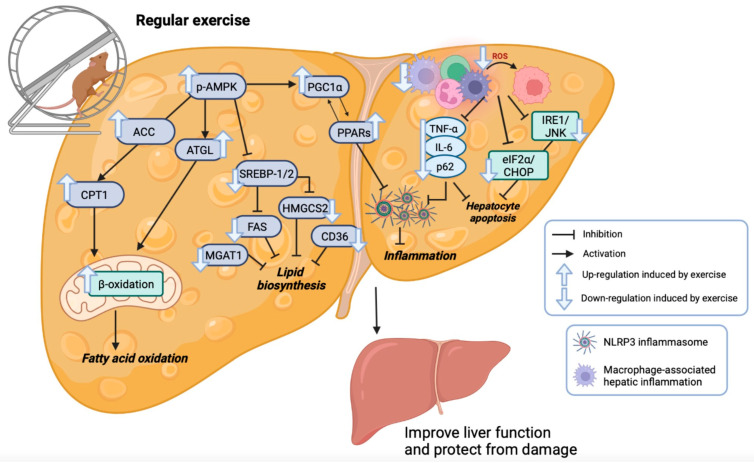
Effects of regular exercise training on metabolic pathways in an animal model of fatty liver. created by BioRender.com. List of abbreviations: Acc: acetil-CoA carboxilasa; Atgl: adipose triglyceride lipase; CHOP: C/EBP homologous protein; Cpt-1: carnitine palmitoyltransferase-1; eIF2α: eukaryotic translation initiation factor 2α; Fas: fatty acid synthase; Hmgcs2: 3-hydroxy-3-methylglutaryl-CoA synthase 2; Il-6: interleukin 6; IRE1: inositol-requiring enzyme 1α; JNK: Jun N-terminal kinases; Mgat1: monoacylglycerol O-acyltransferase 1; NLRP3: NACHT, LRR, and PYD domains containing protein 3; p-Ampk: phosphor-AMP-activated protein kinase; Pgc1α: PPAR coactivator-1 alpha; Ppars (Pppar-α, Ppar-δ and Ppar-γ): peroxisome proliferator-activated receptor α, δ and γ; Srebp1/Srebp2: sterol regulatory element binding transcription factor 1 and 2; TNF-α: tumor necrosis factor alpha.

## Data Availability

Not applicable.

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
