# Peer review of "Physical Activity Protocols in Non-Alcoholic Fatty Liver Disease Management: A Systematic Review of Randomized Clinical Trials and Animal Models"

_healthcare, 2023, doi:10.3390/healthcare11141992_

Round 1

Reviewer 1 Report

This study discusses the possible mechanisms and efficacy of exercise in experimental and clinical NAFLD.

1. This topic has been extensively studied before, although with different emphases. It therefore lacks novelty. The author should emphasise which new insights this manuscript offers the reader compared to previous research.

2. Although the authors state that they are not considering studies that combine exercise with other interventions, this is an important current treatment and prevention tool for NAFLD. Therefore, this section may also need to be described to illustrate the independent/combined effects between exercise and other lifestyle interventions. Alternatively, provide an explanation as to why only exercise-related studies were included.

3. The tables seem to be images and are very unclear, please provide editable tables rather than images.

4. What is the best current exercise protocol, and what are the official guidelines or expert opinions in this area.

5. Quality assessment of the included studies was not provided.

6. Although a section on strengths and weaknesses is provided, more discussion is lacking, as well as an outlook on future research directions.

7. Whether the improvement of extrahepatic complications by exercise is described in the literature. Whether the compliance and effectiveness of exercise differs between NAFLD severities.

Minor editing of English language required.

Author Response

Reviewer 1.

This study discusses the possible mechanisms and efficacy of exercise in experimental and clinical NAFLD.

  1. This topic has been extensively studied before, although with different emphases. It therefore lacks novelty. The author should emphasize which new insights this manuscript offers the reader compared to previous research.

Answer: We appreciate the reviewer's comment. We consider that the novelty that this review provides is the inclusion of both animal models and human clinical trials studies, which allows us to compare and show which was the strategy with the greatest scientific evidence of exercise training without other types of intervention for the prevention and treatment of NAFLD. Thus, we highlight the novelty of this manuscript on Page 2, lines 80 to 84.

  1. Although the authors state that they are not considering studies that combine exercise with other interventions, this is an important current treatment and prevention tool for NAFLD. Therefore, this section may also need to be described to illustrate the independent/combined effects between exercise and other lifestyle interventions. Alternatively, provide an explanation as to why only exercise-related studies were included.

Answer: We included only exercise interventions because, despite being a strategy known to have multiple health benefits, the specific training prescription is still not clear and its promotion in clinical practice is not enough. In this review we tried to highlight that exercise is the lifestyle changes with the most benefits in this pathology, and to mention the main metabolic and molecular pathways involved in exercise training. This information was included on page 2, lines 75-84.

  1. The tables seem to be images and are very unclear, please provide editable tables rather than images.

Answer: We appreciate the reviewer's comment, the editable tables and images have already been included in the document.

  1. What is the best current exercise protocol, and what are the official guidelines or expert opinions in this area.

Answer: The exercise prescription in terms of type, intensity, frequency, and duration has not been standardized. However, the American College of Sports Medicine (ACSM) recommends at least 150 min/week of moderate or 75 min/week of vigorous-intensity physical activity for all patients with NAFLD. We appreciate the reviewer's question, and we included this information on pages 2, lines 75-80

  1. Quality assessment of the included studies was not provided.

Answer: We appreciate the reviewer's observation, the quality assessment was included on page 5, lines 241-245, and we also included the figure 2.

  1. Although a section on strengths and weaknesses is provided, more discussion is lacking, as well as an outlook on future research directions.

Answer: This observation about future research directions was addressed in Strengths and Limitations section, page 29, lines 650-656

  1. Whether the improvement of extrahepatic complications by exercise is described in the literature. Whether the compliance and effectiveness of exercise differs between NAFLD severities.

Answer: We appreciate the reviewer's criteria. This point was included on Strengths and Limitations section, page 29, lines 639-641.

Reviewer 2 Report

My comments to the authors are as follow.

Title:

It's better to say "a systematic review of randomized clinical trials" because the review is limited to RCTs.

Also, it's better to include human and mice models in the title if it is possible.

Keywords:

The keywords of "Exercise; NAFLD; NASH" are not good, please replace them with proper keywords. You can use PubMed MeSH database.

Introduction:

- "Currently, there is no approved effective pharmacological treatment for NAFLD." This statement is too general. By "approved", do you mean FDA approved? Please specify and provide a reference for it.

- The following sentence is about NASH while the rest of paragraph is about NAFLD. Please move this sentence to a better place.

"The American Gastroenterological Association (AGA) advises patients with NASH to reduce over 5 % of total 51 body weight (TBW) to decrease hepatic steatosis, over 7 % for NASH resolution, and over 52 10 % to resolve/stabilize fibrosis, as has been shown in several studies [8–11]."

- "A sedentary lifestyle has been linked to a detrimental prognosis in individuals with non-alcoholic fatty liver disease (NAFLD)". You have used the NAFLD abbreviation before.

- "In this context, programmed exercise has a role in the prevention and therapeutic intervention at a low cost for patients." What do you mean by low cost? This sentence needs cost-effective studies. Provide reference for this sentence or revise it.

Methods:

- In the search strategy, I think “NASH” is not a proper keyword. You should also use the extended form.

- "The search was restricted to studies published from January 2017 to November 2022." Is there any reason for limiting the publication date?

- In figure 1a, 5 review articles excluded during the full-text screening? These should be excluded in title or abstract screening.

- In figure 1a and 1b, the reasons for excluding articles after full-text screening need more explanation. What do you mean by "no supervised exercise training", "insufficient...", "alternative...", "use of drug...", "other reasons", etc. Please explain these criteria in the text.

Results:

- Tables are low-quality pictures. I cannot see some parts of the tables.

- I could not find a legend and list of abbreviations for figure 2.

Discussion:

- "Our review was carefully revised addressing that only RCTs were included, which deliver the highest level of evidence, thus preventing selection bias." RCT is not the highest level of evidence. Please revise this sentence.

- Although limiting studies to RCTs is a strength point, it can also be a limitation. You excluded the big cohort studies.

- Another limitation about this review is heterogenicity between the studies regarding the study population, type of intervention, and outcome assessment. The authors also did not evaluate the effect of race and diet.

Author Response

RESPONSE TO REVIEWERS

Reviewer 2.

Title:

It's better to say "a systematic review of randomized clinical trials" because the review is limited to RCTs.

Also, it's better to include human and mice models in the title if it is possible.

Answer: We appreciate the reviewer's comment. Therefore, we modified the title to include this suggestion: Physical Activity Protocols in Non-Alcoholic Fatty Liver Disease Management: A Systematic review of randomized clinical trials and animal models. Page 1, lines 2-4.

Keywords:

The keywords of "Exercise; NAFLD; NASH" are not good, please replace them with proper keywords. You can use PubMed MeSH database.

Answer: We attended the suggestion, and we changed the words to: Physical activity; Exercise training; Liver diseases, Fatty liver according to PubMed MeSH database. Page 1, line 33

Introduction:

- "Currently, there is no approved effective pharmacological treatment for NAFLD." This statement is too general. By "approved", do you mean FDA approved? Please specify and provide a reference for it.

Answer: Currently, no drug has been approved by international or local regulatory agencies specifically to treat NAFLD. However, there are many drugs approved to treat other components of metabolic syndrome such as diabetes mellitus, dyslipidemia, or obesity for example. We appreciate the comment, and the statement was cleared on page 2, lines 49-51, and added the following reference:

David D, Eapen CE. What Are the Current Pharmacological Therapies for Nonalcoholic Fatty Liver Disease? J Clin Exp Hepatol. 2021 Mar-Apr;11(2):232-238. doi: 10.1016/j.jceh.2020.09.001. Epub 2020 Sep 3. PMID: 33746449; PMCID: PMC7953000.

- The following sentence is about NASH while the rest of paragraph is about NAFLD. Please move this sentence to a better place.

"The American Gastroenterological Association (AGA) advises patients with NASH to reduce over 5 % of total 51 body weight (TBW) to decrease hepatic steatosis, over 7 % for NASH resolution, and over 52 10 % to resolve/stabilize fibrosis, as has been shown in several studies [8–11]."

Answer: We restructured the writing to order the information more clearly (Page 2, Line 54 and 55)

- "A sedentary lifestyle has been linked to a detrimental prognosis in individuals with non-alcoholic fatty liver disease (NAFLD)". You have used the NAFLD abbreviation before.

Answer: We appreciate the observation and we have corrected this mistake and used the abbreviation correctly (Page 2, line 65)

- "In this context, programmed exercise has a role in the prevention and therapeutic intervention at a low cost for patients." What do you mean by low cost? This sentence needs cost-effective studies. Provide reference for this sentence or revise it.

Answer: The low cost refers to the fact that the practice of physical exercise does not always represent an important economic expense compared to the chronic consumption of drugs. Since there are several options to practice exercise, including rutines from home, parks, and public places, it is a type of low-cost therapy. In this context, lifestyle modifications such as increase physical activity and exercise training has low nominal cost relative to long drug-based treatments and the natural history of this pathology.

 We provided a better sentence to clarify the message and we appreciate the remark (Page 2, line 69-72). We also added the following references:

Rustgi, V. K., Duff, S. B., & Elsaid, M. I. (2022). Cost-effectiveness and potential value of pharmaceutical treatment of nonalcoholic fatty liver disease. Journal of medical economics, 25(1), 347–355. https://doi.org/10.1080/13696998.2022.2026702

Rushing J, Wing R, Wadden TA, Knowler WC, Lawlor M, Evans M, Killean T, Montez M, Espeland MA, Zhang P; Look AHEAD Research Group. Cost of intervention delivery in a lifestyle weight loss trial in type 2 diabetes: results from the Look AHEAD clinical trial. Obes Sci Pract. 2017 Mar;3(1):15-24. doi: 10.1002/osp4.92. Epub 2017 Feb 24. PMID: 28392928; PMCID: PMC5358076.

Methods:

- In the search strategy, I think “NASH” is not a proper keyword. You should also use the extended form.

Answer: This observation was included as suggested by the reviewer (Page 5, line 215-216)

- "The search was restricted to studies published from January 2017 to November 2022." Is there any reason for limiting the publication date?

Answer: The date was restricted to the last 5 years according to publication date to include the most updated information on this topic. (Page 5, Line 218).

- In figure 1a, 5 review articles excluded during the full-text screening? These should be excluded in title or abstract screening.

Answer: We support this comment, and the review articles were excluded from the title/abstract reviewing. The flowchart 1a was corrected (page 3).

- In figure 1a and 1b, the reasons for excluding articles after full-text screening need more explanation. What do you mean by "no supervised exercise training", "insufficient...", "alternative...", "use of drug...", "other reasons", etc. Please explain these criteria in the text.

Answer: We appreciate the reviewer's comments, and this suggestion has been explained on detail on page 5, lines 230-234.

Results:

- Tables are low-quality pictures. I cannot see some parts of the tables.

Answer: We appreciate the reviewer's comment; the editable tables have already been included in the document

- I could not find a legend and list of abbreviations for figure 2.

Answer: We appreciate the reviewer's comment, the legend and list of abbreviations has already been included in the figure 2 (now figure 3, page 21, lines 434-439).

Discussion:

- "Our review was carefully revised addressing that only RCTs were included, which deliver the highest level of evidence, thus preventing selection bias." RCT is not the highest level of evidence. Please revise this sentence.

Answer: We appreciate the comment, and the sentence has been correctly written on page 29, line 632.

- Although limiting studies to RCTs is a strength point, it can also be a limitation. You excluded the big cohort studies.

Answer: We appreciate the observation reviewer’s, and we included this limitation in the main document on page 29, line 638.

- Another limitation about this review is heterogenicity between the studies regarding the study population, type of intervention, and outcome assessment. The authors also did not evaluate the effect of race and diet.

Answer: We appreciate this observation. We did not consider diet due to the studies selected for this review: nutritional interventions were an elimination criterion because we were focused only on the benefits of the exercise protocol in NAFLD. It was also not possible to consider ethnicity since the number of clinical exercise interventions are few, so we decided to avoid it as elimination criteria to include the largest number of studies possible. Therefore, we addes more discussion of this point on page 29, lines 638-641.

Round 2

Reviewer 1 Report

The author has addressed the issue relatively well. Acceptable.

Reviewer 2 Report

Thank you for giving me the opportunity to review the revised version of this manuscript. While the title may not be groundbreaking and the study does possess certain limitations, I believe that this revised version is well-suited for publication in the Healthcare journal.

Best regards